# Characterization of Human Transition Zone Reveals a Putative Progenitor-Enriched Niche of Corneal Endothelium

**DOI:** 10.3390/cells8101244

**Published:** 2019-10-12

**Authors:** Gary Hin-Fai Yam, Xinyi Seah, Nur Zahirah Binte M Yusoff, Melina Setiawan, Stephen Wahlig, Hla Myint Htoon, Gary S.L. Peh, Viridiana Kocaba, Jodhbir S Mehta

**Affiliations:** 1Tissue Engineering and Stem Cell Group, Singapore Eye Research Institute, Singapore 169856, Singapore; 2Eye-Academic Clinical Program, Duke-National University of Singapore (NUS) Graduate Medical School, Singapore 169857, Singapore; 3Department of Ophthalmology, Duke University School of Medicine, Durham, NC 27705, USA; 4Data Science Group, Singapore Eye Research Institute, Singapore 169856, Singapore; 5Department of Ophthalmology, Claude Bernard Lyon 1 Université, 69622 Villeurbanne, France; 6Singapore National Eye Centre, Singapore, Singapore 168751, Singapore; 7School of Material Science and Engineering, Nanyang Technological University, Singapore 639798, Singapore

**Keywords:** corneal endothelium, posterior limbus, transition zone, corneal endothelial progenitors, morphology, cell culture

## Abstract

The corneal endothelium regulates corneal hydration to maintain the transparency of cornea. Lacking regenerative capacity, corneal endothelial cell loss due to aging and diseases can lead to corneal edema and vision loss. There is limited information on the existence of corneal endothelial progenitors. We conducted ultrastructural examinations and expression analyses on the human transition zone (TZ) at the posterior limbus of corneal periphery, to elucidate if the TZ harbored progenitor-like cells, and to reveal their niche characteristics. Within the narrow TZ (~190 μm width), the inner TZ—adjacent to the peripheral endothelium (PE)—contained cells expressing stem/progenitor markers (Sox2, Lgr5, CD34, Pitx2, telomerase). They were located on the inner TZ surface and in its underlying stroma. Lgr5 positive cells projected as multicellular clusters into the PE. Under transmission electron microscopy and serial block face-scanning electron microscopy and three-dimensional (3D) reconstruction, the terminal margin of Descemet’s membrane was inserted beneath the TZ surface, with the distance akin to the inner TZ breadth. Porcine TZ cells were isolated and proliferated into a confluent monolayer and differentiated to cells expressing corneal endothelial markers (ZO1, Na^+^K^+^ATPase) on cell surface. In conclusion, we have identified a novel inner TZ containing progenitor-like cells, which could serve the regenerative potential for corneal endothelium.

## 1. Introduction

The corneal endothelium (CE) is a monolayer of cells covering the posterior corneal surface and is crucial for corneal transparency [1]. The hexagonal-shaped corneal endothelial cells (CECs) are tightly packed and arranged in a tessellated configuration on their basement membrane, the Descemet’s membrane (DM), forming a barrier with simple diffusion. Mature CECs are metabolically active, with continuous ATPase activity for the fluid-coupled active transport of ions from the corneal stroma to aqueous humor [2]. This “leaky barrier and ionic pump” activity regulates the stromal hydration and prevents edema in maintaining the corneal transparency, which is necessary for normal vision [3,4,5]. Under optical coherence tomography, it is evident that the CE terminates at the Schwalbe’s line in the corneal periphery [6]. Under scanning electron microscopy (SEM) of higher magnification, there lies a thin strip of tissue—the transition zone (TZ)—also known as the flat or smooth zone, or Zone S [7,8]. The TZ has been previously described as a smooth annular region devoid of CECs and trabecular meshwork (TM) fibers [9]. The inner border of TZ is adjacent to the peripheral endothelium (PE), and the outer border is demarcated by the anterior non-filtering portion of TM with beam inserts and bridges [7]. However, the tissue anatomy of TZ is poorly characterized.

The human CE is known to have very limited regenerative capacity [10]. Mature CECs are non-mitotic due to their expression of negative cell cycle regulators (including CIP, INK and p53 protein families), contact inhibition and the presence of mitogenic inhibitors (such as transforming growth factor-β, TGFβ) in aqueous humor [11]. Hence the gradual loss through aging, and the traumatic damage from diseases or injury lead to the CE dysfunction [12]. Over the last decade, there have been repeated attempts to look for putative CE progenitors. During the development of anterior segment, neuroectoderm-derived cranial neural crest cells (NCCs) migrate from the neural tube to form the optic vesicle. A loose array of cranial NCCs, termed as periocular mesenchyme (POM), derive multiple cell types in the anterior segment, including CE, stroma, TM, Schlemm’s canal, ciliary body and iris stroma [13]. Our group recently identified cells co-expressing HNK1 and p75^NTR^ (NCC markers) and Pitx2 (POM marker), that resembled NCC/POM cells, inside the adult human TZ [14]. Previous reports on the expression of nestin, alkaline phosphatase and telomerase (TERT), and injury-induced Wnt-1 signaling have suggested the existence of progenitors in the corneal periphery [15,16]. Human PE cells have been shown to form spheres under non-adherent culture, demonstrating the proliferative potential [17,18,19]. Likewise, endothelial cells from the paracentral cornea were able to demonstrate neuroplasticity [20]. Hence, progenitor-like cells could exist in the posterior limbus; however, their exact location is unclear.

Although TZ has been identified [7,8], it is still poorly characterized and is uncertain if putative progenitors are sequestered in specific niches inside the TZ and/or PE, and the synergy between progenitors and niche. A previous microanatomical study has revealed multilayered cell clusters and radial cell rows in PE, suggesting a possible migration of less differentiated cells from the corneal periphery to the center [21]. Though PE cells could have limited proliferative capacity, they may hold a potential to divide ex vivo under conditions without mitogenic inhibitors. In this study, we have examined whether PE is a primary progenitor niche or whether PE cells are originated from the neighboring TZ. This would provide important information on the localization of endothelial progenitors.

Here, we present a comprehensive characterization study of human TZ tissue using stem cell marker expression in corroboration with scanning, transmission, and 3D-EM to elucidate the existence of progenitor-like cells and the niche anatomy. The results were compared to the adjacent PE and TM. The differentiation potential of porcine TZ progenitors to form CE-like cell monolayer was verified by the expression of standard CE markers.

## 2. Materials and Methods

### 2.1. Human Corneas

Clinical grade cadaveric corneal tissues (*n* = 53) from 47 donors (male/female: 24/23; age: 49.4 ± 15.8 years old; age range: 18 to 76 years old) (Appendix A) were procured from Lions Eye Institute for Transplant and Research Inc. (Tampa, FL, USA) and Lyon Cornea Eye Bank (Edouard Herriot Hospital, Hospices Civils de Lyon, Lyon, France) with consent for clinical and research use taken at the time of retrieval by the next of kin. The study protocol was approved by the Centralized Institutional Research Board, SingHealth, Singapore (2015/2320) and carried out in accordance with the tenets of the Declaration of Helsinki. Corneal tissues were transported in Optisol-GS (Bausch & Lomb, Bridgewater, NJ, USA) at 4 °C. For orientation-marked corneas, eye bank technicians were specially requested to mark the corneas at the most nasal position with a scleral notch, at the time of enucleation.

### 2.2. Whole Mount Histochemistry and Immunostaining

Corneal rims devoid of iris tissue were fixed in 2% paraformaldehyde (Sigma-Aldrich). The rinsed samples were saponin-permeabilized and blocked with bovine serum albumin (BSA, 2%, Sigma-Aldrich) and normal goat serum (5%, ThermoFisher, Waltham, MA, USA), followed by incubation with or without primary antibodies or host species-matched isotype-specific immunoglobulin (Ig) (Appendix A) overnight at 4 °C. After washes, they were stained with appropriate AlexaFluor 488 or AlexaFluor 594-conjugated IgG/IgM secondary antibody (Jackson ImmunoRes Lab, West Grove, PA, USA) and/or phalloidin-fluorescein conjugate (Invitrogen), washed and mounted with Fluoroshield containing 4’,6-diamidino-2-phenylindole (DAPI) (Santa Cruz Biotech, Santa Cruz, CA, USA). Alternatively, the corneal rim samples were cryo-embedded in optimal cutting temperature (OCT) compound (Tissue-Tek, VWR, Singapore) and sectioned (6 µm thick). Immunostaining was performed as before, followed by fluorescence-conjugated secondary antibody. Serial z-stack images (1 μm thickness) were collected by laser-scanning confocal microscopy (TCP SP8, Leica, Wetzlar, Germany; AxioImager II, Carl Zeiss) and 3D-reconstructed montaged images were obtained using LAS X software (Leica, Wetzlar, Germany). The staining intensity profiles were analyzed using ImageJ software (Fiji version, National Institute of Health, Bethesda, USA) after the antibody-specific channel was grey-scaled and thresholded to background level. A total of 3 samples with a minimum of 6 fields for each antibody-stained image were analyzed.

### 2.3. Scanning Electron Microscopy (SEM) and TZ Width Measurement

Orientation-marked human corneal rims (*n* = 5; donor age: 61.8 ± 10.6 years old) were fixed in 3% glutaraldehyde (EM Sciences, Hatfield, PA, USA) in 0.1 M sodium cacodylate buffer (pH 7.5, Sigma-Aldrich) for 2 h. Each rim was cut into 8 equal pieces (arc length of 4.5 to 4.7 mm) (Figure 1A) and labeled according to the orientation. They were post-fixed in 1% aqueous osmium tetroxide (OsO_4_, EM Sciences), dehydrated, critical point dried, and sputter-coated with gold alloy (10 nm thick). TZ images were collected using FEI Quanta 650 FEG SEM (JEOL, Tokyo, Japan) at 300× magnification. TZ was outlined anteriorly at the border of PE and posteriorly at the uveal insertions into TM (Figure 1C). At every 100 µm interval, a line was drawn between borders, and the length measured using the ruler tool of Photoshop CS with reference to the calibrated scale bar. The measurement was done by X.S. after method validation, described in the next section. Another group of 17 corneal rims without orientation marks were similarly processed for mean TZ width analysis (Appendix A).

The method of TZ width measurement was validated through inter- and intra-observer agreements. Two observers (G.H.-F.Y., S.W.) independently performed width measurement on 4 corneal rims, each had 16 equal pieces (a total of 64 data points), using the ruler tool of Photoshop CS and the length measurement tool of ImageJ, respectively. The exercise was repeated after 48 h and observers were masked to the earlier results. We compared inter- and intra-observer reproducibility using Bland-Altman plot and *p* < 0.05 indicated the significant variability.

### 2.4. Quantitative Reverse Transcription-Polymerase Chain Reaction (qRT-PCR)

From human cornea (*n* = 11), central CE was collected after DM peeling (central 8 mm diameter). The posterior limbus was collected by an inward peeling method (https://www.youtube.com/watch?v=wQlPhHCJgmM&t=121s). Under stereo-microscope, the tissue was further dissected to separate the innermost transparent PE with <0.5 mm width, the outermost pigmented TM tissue, and the remaining intermediate non-pigmented smooth TZ. All samples were washed at least three times in ice-cold PBS, followed by preservation in Trizol reagent (Sigma-Aldrich). Total RNA was extracted using RNeasy kit (Qiagen, Hilden, Germany) and on-column RNase-free DNase kit (Qiagen). After reverse transcription using Superscript III RT-PCR kit (ThermoFisher), cDNA was assayed for candidate gene expression with specific primer pairs (Appendix A) by quantitative real-time PCR (qPCR) using Sybr Green Supermix (BioRad, Herculus, CA, USA) or Taqman assay in GFX96 real-time system (BioRad). Experiments were run in quadruplicate. The relative gene expression (ΔCT) normalized with mean CT of glyceraldehyde-3-phosphate dehydrogenase (GAPDH) (CT_GAPDH_) or β-actin (ACTB) (CT_ACTB_) was obtained, and fold change to the central CE was expressed as mean ± SD.

### 2.5. Transmission Electron Microscopy (TEM)

Human corneal rims (*n* = 4) were fixed in 3% glutaraldehyde in sodium cacodylate buffer for 2 h at 4 °C. The rinsed samples were post-fixed in 1% aqueous OsO_4_, processed for Epon-Aradite infiltration and embedding, and ultrathin sectioning (90 nm thick). After contrast staining with 3% uranyl acetate (EM Sciences) and lead citrate, sections were examined under TEM (JEOL 2100).

### 2.6. Serial Block-Face Scanning Electron Microscopy (SBF-SEM)

Human corneal rims (*n* = 2) were fixed in 2.5% glutaraldehyde/4% paraformaldehyde in sodium cacodylate buffer for 2 h at 4 °C and cut into quadrants. The washed samples were stained in 2% OsO_4_/1.5% ferrocyanide for 2 h at 4 °C, followed by 1% tetracarbohydrazide at 60 °C and 2% aqueous OsO_4_ for 2 h at RT. Samples were stained en block in saturated aqueous uranyl acetate for 24 h, then in Walton’s lead aspartate for 45 min at 60 °C. The rinsed samples were dehydrated, cleared in propylene oxide, and embedded in Epon812 resin (EM Sciences). The sample blocks were mounted in a Zeiss Sigma VP SEM, equipped with a Gatan 3View in-chamber ultramicrotome and operated under low kV setting for SBF imaging (performed by Renovo Neural Inc., Cleveland, OH, USA). The acquired z-series images were batch-processed and aligned. Manual segmentation of TZ cells, TZ surface, PE cells and DM were done with Imaris 9.2 (BitPlane AG, Singapore) on each slice with voxel size 0.0151 × 0.0151 × 0.130 μm. Each component was 3D reconstructed using surface generation.

### 2.7. Porcine TZ Cell Culture

Fresh porcine eyes (*n* = 6) from a local abattoir were sterilized with providone iodine (Prodine, ICM Pharma, Singapore). The posterior limbus was isolated by inward peeling method, and TZ (without pigmented TM and translucent PE) was digested with collagenase I (0.2%, Worthington, Lakewood, NJ, USA) for 3 h at 37 °C. The cell suspension was passed through 40 μm cell strainer (Falcon) and single cells were seeded on matrigel-coated culture surface in TZ stem cell medium (TZSCM), which was OptiMEM1 with recombinant human epidermal growth factor (h-EGF, 10 ng/mL, ThermoFisher), recombinant human basic fibroblast growth factor (h-bFGF, 20 ng/mL, ThermoFisher), bovine pituitary extract (100 μg/mL, Sigma-Aldrich), L-ascorbate (20 μg/mL), chondroitin sulfate (0.08%), calcium chloride (0.9 mM), and knockout serum replacement (5%, ThermoFisher). After 14 days, primary colonies were collected for CEC culture using a dual media approach [22,23]. Briefly, TZ cells were propagated on culture surface pre-coated with FNC mix (AthenaES, Baltimore, MD, USA) and 0.02% chondroitin sulfate in proliferative M4 medium, which was Ham’s F12/M199 (ThermoFisher) added with Equafetal bovine serum (5%, Atlas Biologicals, Fort Collins, CO, USA), ROCK inhibitor (Y27632, 2 μM, Millipore, Burlington, MA, USA), l-ascorbate (20 μg/mL), and h-bFGF (10 ng/mL). At confluence, the culture was replenished with stabilizing M5 medium, which was human endothelial-serum free medium (ThermoFisher, catalogue number 11111-044) containing h-bFGF (20 ng/mL), h-EGF (10 ng/mL), human plasma fibronectin (10 μg/mL), and 5% Equafetal bovine serum for 7 to 10 days. Cells were collected for CE marker expression analyses.

### 2.8. Western Blotting

Cells were lyzed in a radioimmunoprecipitation assay (RIPA) buffer (ThermoFisher) added with Complete^TM^ protease inhibitor cocktail (Roche, Basel, Switzerland) and phenylmethylsulfonylfluoride (PMSF, 1 mM, Sigma). Soluble proteins were denatured in sodium dodecylsulfate (SDS, 2%) loading buffer with dl-dithiothreitol (200 mM, Sigma-Aldrich), resolved by 4–20% SDS-polyacrylamide gel electrophoresis (SDS-PAGE, BioRad) and transferred to nitrocellulose membrane (BioRad). After blocking with 5% nonfat milk, the membrane was incubated with primary antibodies against Prdx6, Na^+^K^+^ATPase, ZO-1 and housekeeping β-actin (Appendix A), followed by horseradish peroxidase-conjugated Ig antibodies (Roche). Staining signals were revealed by enhanced chemiluminescence (ThermoFisher) using a ChemiDoc XRS gel imaging system (BioRad). Band densitometry was conducted using QuantityOne 1D analysis (BioRad). Specific band density was normalized to that of β-actin. Experiments were conducted in triplicate and results were represented as mean ± SD.

### 2.9. Statistics

Bland–Altman plot analysis (MedCalc, v.9.6.4.0, Ostend, Belgium) was used for inter- and intra-observer agreement on TZ width measurement. The reproducibility was calculated in terms of limits of agreement (LOAs) (mean of differences ± 1.96 SD). The difference between observers was determined by paired t-test. Correlation of TZ width with donor age and gender was assessed by Mann–Whitney U test and Spearmen’s test, respectively. Mann–Whitney U test was also used to calculate the significance of gene expression difference between samples. *p* < 0.05 was considered statistically significant.

## 3. Results

### 3.1. Posterior Limbus Contained Anatomically Distinguishable Regions

The human posterior corneal periphery possessed regions that marked a transition from the iris root to corneal endothelium (CE) (Figure 1A,B; Appendix A). Adjacent to the iris root, the wedge-shaped band of TM contained insertions anastomosed randomly to form a meshwork. The relatively flattened region of TZ had its outer border in continuity with TM (Figure 1B). The cells on the TZ surface, facing the anterior chamber, exhibited irregular sizes and polygonal cell shape, when compared to the adjacent peripheral endothelium (PE) with regularly arranged tightly packed hexagonal cells (Figure 1C). Alizarin red S staining (Appendix A) revealed that PE cell borders had more intense staining than cells located in the TZ, illustrating tighter cell-cell arrangement in PE than that in TZ (Appendix A). In 5 corneal samples (with CEC count 2602 ± 261 cells/mm^2^), the cell density revealed by DAPI staining was significantly lower in TZ (708 ± 134 cells/mm^2^; range 564 to 845 cells/mm^2^) than in the PE immediately adjacent to TZ (3125 ± 206 cells/mm^2^; range 2872 to 3334 cells/mm^2^) (*p* = 0.012, Mann–Whitney U test) (Appendix A).

### 3.2. TZ Width Measurement and Distribution in Human Corneas

We first validated the TZ width measurement method using montaged SEM images of 4 corneal rims. The outer and inner TZ borders were outlined as described (Figure 1C). Two masked observers independently measured TZ width in a total of 64 sites. Bland–Altman plot analysis showed good inter- and intra-observer agreements (Figure 1D and Appendix A). The levels of agreement (LOAs) were comparatively low with a mean difference <0.69 µm for inter-observers and <0.62 µm for intra-observers. Although one statistically significant bias was seen for TZ65 (*p* = 0.04), we showed strong overall agreement for inter- and intra-observer measurements of TZ width at standard points and validated the high reproducibility of our method, which was used in determining TZ width in following experiments.

The mean TZ width of 17 corneas (donor age: 52.4 ± 8.9 years old; range: 27 to 70 years old; male/female: 1.3:1; Appendix A) with fully measurable 360° profiles was 191 ± 50 µm (from an average of 319 measurements per sample). The individual TZ widths had highly variable distribution (from 33 to 700 μm) and the median TZ width was in the range of 125 to 150 µm, which had 15% coverage among all measurements (Figure 1F). Moreover, 69% of TZ width was in the range of 100 to 250 µm. The demographic information of donors and its association with TZ width are depicted in Appendix A. The mean TZ width of Group 1 (≤60 years old, *n* = 6) was 169.7 ± 35.7 μm and Group 2 (>60 years old, *n* = 11) was 203.2 ± 55.8 μm. There was no significant difference concerning the mean TZ width distribution between these two age groups (*p* = 0.246; Mann–Whitney U test) (Appendix A). Donor gender was also not associated with TZ width (Appendix A).

The TZ width distribution with respect to eye orientation was also assessed on corneas marked for orientation (*n* = 5 corneas). Measurement done on SEM images of equal-sized quadrants showed that the nasal quadrant had the narrowest mean TZ width (Figure 1E). Compared to the superior quadrant (set as 1), the mean TZ width ratio of nasal quadrant was 0.81 ± 0.08, inferior was 0.94 ± 0.12 and temporal was 0.93 ± 0.25. Pairwise non-parametric Mann–Whitney U test showed that the nasal TZ was significantly thinner than that in the superior quadrant (*p* = 0.012), but not for inferior and temporal TZ (*p* = 0.144 for inferior and *p* = 0.529 for temporal).

### 3.3. Stem Cell/Progenitor Gene Expression in TZ

We isolated central endothelium, PE, TZ and TM tissues (Figure 2A) from 11 human corneas (age: 56.2 ± 13 years old; male/female: 6/5) (Appendix A). Without pooling, the samples were subject to qPCR analysis for marker expression. When compared to central endothelium, the pluripotency genes (*OCT3/4*, *C-MYC*, *KLF4*, *ABCG2*, *NANOG*), NC genes (*SNAIL1*, *NESTIN*, *SOX9*) and POM genes (*PITX2*, *FOXC1*, *p75^NTR^*) were expressed stronger in TZ, PE and TM (Figure 2B,D,E). Among the tissues of posterior limbus, TZ and PE had higher pluripotency, NC, and POM gene expression than TM. In contrast, the TZ had the most down-regulated expression of *ZO1*, *SLC4A11* and *COL8A2* (corneal endothelial markers), followed by TM and PE, when compared to central endothelium (Figure 2D).

### 3.4. Phenotypically Distinct Regions Inside TZ

Twelve human corneal rims (age 36.5 ± 14 years old; range 18 to 57 years; male to female ratio 1:1) were used for whole-mount immunostaining, followed by confocal microscopy and z-series reconstruction. Figure 3 shows the representative images of marker expression. With the co-staining of DAPI and phalloidin, PE was identified with tightly packed DAPI-stained nuclei, whereas TM had strong phalloidin staining due to the extensive fibrillar insertions and ridges. In contrast, TZ was distinguished by the relatively dim signals of DAPI and phalloidin. In the majority of samples, PE expressed Lgr5 (Figure 3B and Figure 4B) and Prdx6 (TAG2A12) (Figure 3G), with relatively weak staining of nestin (Figure 3E), CD34 (Figure 3D) and telomerase (TERT) (Figure 3A). In TM, positive immunoreactivity was found for vimentin (Figure 3I), and CD44 (Figure 3H). Immunostaining without primary antibody or host species-matched isotype-specific Ig revealed negligible background staining and non-specificities, respectively (Appendix A).

Inside the TZ, 3 different marker expression patterns were observed.Entire TZ expression: Vimentin and POM markers (HNK1) were expressed in the entire TZ (Figure 3F,I). While vimentin was also detected in TM and PE, HNK1 was predominantly found in TZ. On cryo-sections of posterior limbus, Pitx2 (POM marker) was distinctly expressed in the entire TZ, with signals colocalized with DAPI-stained nuclei (Figure 4A). The positive cells were located on the TZ surface and in the immediate stromal region. Minimal Pitx2 signal was detected in the TM region.Inner TZ expression: A number of stem/progenitor markers (TERT, Sox2, CD34, nestin) were expressed in the inner TZ (immediately adjacent to PE) (Figure 3A,C,D,E). The signal profiles showed that intensities were stronger in the inner TZ than in the outer TZ (next to TM), TM and PE regions. Moreover, Lgr5 and Prdx6 (TAG2A12) were detected in the inner TZ and PE, but not in the outer TZ (Figure 3B,G). On cryo-sections of posterior limbus, Lgr5 was distinctly expressed in inner TZ and PE (Figure 4B), of which the cells showed the surface Lgr5 labeling (inset in Figure 4B). We confirmed this result with silver-enhanced immunogold scanning EM (Appendix A). Clusters of electron opaque dots detected under secondary electron mode were brightened when shifted to back-scattered electron imaging mode, representing Lgr5 labeling. We observed cell surface Lgr5 expression on cells inside the inner TZ and PE, but not in the outer TZ. Similarly, TERT was clearly expressed in the inner TZ with positive signals found in cells on the TZ surface and in the immediate stromal region (Figure 4C). The staining of CD34, a marker of diverse progenitors and quiescent corneal stromal keratocytes [24], was visualized in some inner TZ cells. They were different to corneal stromal keratocytes, which co-expressed both CD34 and aldehyde dehydrogenase 3A1 (ALDH3A1, a keratocyte marker) (Figure 4D).Outer TZ expression: We detected CD44 expression (mesenchymal stem cell marker) in the outer TZ (Figure 3H).

The regional expression for stem/progenitor and corneal endothelial markers obtained by both RNA expression and immunofluorescence analyses are summarized in Table 1.

### 3.5. Projection of Progenitors from Transition Zone to Peripheral Endothelium

We examined Lgr5 and Pitx2 expression on whole mount corneal rim samples using serial z-stacked confocal imaging and 3D reconstruction. On a horizontal view tilted at 30°, Lgr5 signals were detected predominantly in the inner TZ and faintly in the outer TZ (Figure 5A). Some signals were detected in the TM. The picture with higher magnification clearly showed that cells with positive cell surface Lgr5 staining were located at the margin of the inner TZ adjacent to PE (Figure 5B and arrows). In contrast, Pitx2 was expressed in the entire TZ. Towards the PE region, Lgr5 positive cells were found to extend in a form of elongated cell clusters into the PE (Figure 5C). The length of these cell cluster extensions varied within and among corneal samples. They could reach up to 200 µm from the PE/TZ margin. In contrast, Pitx2 signal gradually reduced inside the PE, and only some sporadic signals were observed.

### 3.6. Ultrastructural Morphology of Transition Zone

Under transmission EM, the posterior limbus possessed a spongy structure facing to the anterior chamber. The top picture of Figure 6 showed a low magnification image of posterior limbus with PE terminus marked by the end of DM (arrow). In the adjacent TZ, the regions of inner and outer TZ were visualized for the anatomical features in greater details. In the inner TZ (Figure 6A–D), the superficial region contained cells located over a loosely arranged stromal matrix (Figure 6A,B). The cell-matrix alignment was irregular and contained numerous interstitial spaces (IS). The cells had a high nuclear/cytoplasmic ratio and the nuclei had loose chromatin (Figure 6D) and pronounced nucleoli (white asterisk in Figure 6C). Cell processes were commonly observed and extended between the loosely arranged collagen lamellae (white arrows in Figure 6B–D). In contrast, the outer TZ (Figure 6E–H) contained non-pigmented and pigmented cells lying in a closely packed matrix (Figure 6E,F). The cells contained more cytoplasmic granules with pigmentation (Figure 6F,G). Some non-pigmented cells also contained cytoplasmic granules and were highly convoluted in shape (dark asterisk in Figure 6G). Occasionally, cells having phagosome structures with irregular deposits (inset in Figure 6H) were observed in the outer TZ.

### 3.7. Termination of Descemet’s Membrane Beneath the TZ Surface

The PE/TZ junction was anatomically examined using low magnification serial block-face scanning EM (SBF-SEM) and a representative transverse section is shown in Figure 7A. The 3D reconstructed image revealed TZ surface cells (green-colored) overlying on a basement membrane (purple-colored) (Figure 7B,C). This TZ basement membrane extended towards the PE and partially covered DM (blue-colored). Likewise, when the DM reached the meeting margin of TZ, it terminated by tapering and inserting below the TZ surface. The distance of DM insert below TZ was measured in 32 random transverse sections and was found to extend to 35.2 ± 5.4% of the entire TZ width. PE cells coded in brown color were located on the DM and some were closely interacting with TZ cells at the margin (Figure 7B,D) (Appendix A. At the margin of DM, a Hassall-Henle structure, which is known as Descemet excrescent, was observed (arrows in Figure 7A–C).

### 3.8. Primary Culture of Porcine TZ Cells to Corneal Endothelial Cells

Porcine TZ was collected without pigmented TM and translucent PE tissues (Figure 8A) for single cell isolation using collagenase. In the primary culture using TZ stem cell medium (TZSCM), clusters of TZ cells appeared within 3 to 4 days after cell seeding. After 14 days, primary cell colonies were collected (Figure 8B,C). They contained cells with extensive cell processes (Figure 8C). The subsequent proliferative M4 culture gave rise to stellate cells (Figure 8D,E). At about 80% confluence (Figure 8F), the culture was switched to endothelial stabilizing M5 medium for 7 days and generated a confluent homogenous monolayer with unique polygonal to hexagonal-like morphology (Figure 8G,H).

By immunostaining, the confluent cells under M5 stabilization expressed ZO1 on cell membrane and Na^+^K^+^ATPase in both cytoplasm and cell surface while the proliferating cells in M4 culture did not (Figure 9A). On the other hand, the nuclear Pitx2 was more synthesized in proliferating cells than the stabilized cells. Western blotting illustrated that ZO1 (195 kDa) and Na^+^K^+^ATPase (105 kDa) were detected in M5 stabilized cells, while Prdx6 (25 kDa) remained unchanged (Figure 9B). Band densitometry results showed the induction of ZO1 by around 4-folds and Na^+^K^+^ATPase by 7–12 folds in confluent cells. The CEC marker induction was replicated in two M5 culture time points (day 7 and 10).

## 4. Discussion

We studied the human TZ ultrastructurally, together with stem/progenitor marker expression, to reveal the presence of progenitor-like cells and their niche characterization. The fully measurable 360° TZ profiles showed the average TZ width was 191 ± 50 µm, with the modal width falling between 125 to 150 μm. Significant TZ narrowing occurred mostly in the nasal quadrant, compared to other regions. No correlation was found between the mean TZ width and donor age of adult corneas and gender. Using immunofluorescence and immunogold SEM, we discovered a progenitor-enriched inner TZ with an exclusive expression of stem/progenitor markers (Sox2, CD34, Lgr5 and TERT). These progenitor-like cells were located on the inner TZ surface and the immediate underlying stroma. Lgr5 positive cells were found to extend from the inner TZ to PE in a digitated form of cell clusters. The 3D-EM results illustrated that, at TZ/PE border, the DM terminus inserted below TZ surface, reaching a distance akin to the breadth of inner TZ. Cultivation of porcine TZ cells using a reported dual media approach generated a CE-like monolayer with CEC-specific expression of ZO1 and Na^+^K^+^ATPase.

Examination of TZ dimension is a primary assessment of its anatomical features, however, no standard method has been reported. Previous calculations using block area of TZ divided by the segment length (~400 μm), were insufficient to account for width variations within the area [8]. Hence, we conducted frequent measurement at every 100 μm interval to reveal the TZ width changes in detail. This method was validated for intra- and inter-observer reproducibility. The measurement results from two independent observers differed insignificantly, illustrating that this is a highly reliable method for TZ width determination.

This method was employed to investigate the fully measurable 360° TZ width profiles of 17 human corneas. The average TZ width was 191 ± 50 μm (ranged from 117 to 292 μm per cornea). This was different to the previous report of 79 ± 22 μm based on block area measurement under scanning EM using formalin-fixed human corneas [8], which might have had tissue shrinkage due to unconventional EM fixation and lack of critical point drying. Another study using donor eyes unsuitable for transplantation reported that TZ widths ranged from 80 to 130 μm [9]. However, no sample size, donor information and methodology were described, hence causing difficulty for further evaluation. Our results showed that the modal width (~15% coverage of all measurements) ranged from 125 to 150 μm. In addition, about 70% of measurements had widths within 100 to 250 μm and this was equivalent to 0.8, ~2%, of corneal diameter (an average of 12.2 mm [25]).

Very few publications have described TZ development, the origin of TZ cells and their possible link to CE progenitors. During development, the periocular mesenchyme (POM), derived from cranial NCC, migrate in three waves to the optic cup [26,27]. The first wave reaching the space between surface ectoderm and lens derives the corneal epithelium and endothelium, the second wave generates stromal keratocytes, and the third wave—towards the anterior chamber angle—contributes to the ciliary body and iris stroma. The POM located anteriorly to the chamber angle between the anterior edge of eyecup and endothelium then develops into cells comprising TM and Schlemm’s canal [28]. As the TZ is located between PE and TM, it presumably follows their developmental path. However, the origin of TZ cells has not been specified: (1) Whether they are derived from the first or the third wave of NCC migration, and (2) whether TZ retains two different populations of POM cells for the respective CE and TM lineage formation. Yu et al. previously proposed that the posterior limbus contained progenitors for endothelium and trabeculum cells with regeneration potential to both CE and TM [9], however, without cell tracing and differentiation studies, the exact cell fate is uncertain.

Total RNA isolated from TZ and PE was found to have greater expression of NC (*SNAIL1*, *NESTIN*, *SOX9*), POM (*PITX2*, *FOXC1*, *p75NTR*) as well as pluripotency genes (*OCT3/4, C-MYC, KLF4, ABCG2, NANOG*) than in CE and TM tissues. Since we were aware of possible contamination of neighboring tissues due to unobvious landmarks during tissue harvest, we examined an increased number of samples (*n* = 11 corneas) to confirm that the elevated stem/progenitor marker expression in TZ and PE was a general phenomenon among corneas. In addition, our cyto-architectural study discovered that the TZ consisted of two phenotypically distinct parts: The inner and outer TZ, and the progenitor-like cells were enriched inside the inner TZ. Various stem/progenitor markers (CD34, Lgr5, Sox2, TERT) were exclusively expressed in the inner TZ, but not in the outer TZ. This novel finding was supported by immunogold SEM. Cells having the surface expression of Lgr5 were populated in the inner TZ, but rarely in outer TZ. Lgr5 positive cells were also found to extend from the inner TZ to the extreme portion of PE in a digitated pattern of multicellular clusters. Earlier reports have shown that extreme PE (within 0.2 mm from TZ border) contained multilayered cell clusters between Hassall-Henle warts and they had greater expression of nestin and TERT, and less of CE differentiation-related ZO1 and Na^+^K^+^ATPase [21]. Hence, our work identified that inner TZ contained progenitors and they could act upstream to the PE cell clusters. CD34, a glycosylated type I transmembrane protein, is expressed in various undifferentiated cell types, including early lymphohaematopoietic stem cells and progenitor cells, embryonic fibroblasts, and dendritic cells in connective tissues [24,29]. This supports our TZ immunostaining result that inner TZ cells expressing CD34 could represent cells associated to primitive phenotype. Their negligible ALDH3A1 expression further suggested that these inner TZ cells could be different to corneal stromal keratocytes, which co-expressed both CD34 and ALDH3A1. However, other reports have shown that CD34 positive cells only represented a subset of stromal cells that derived from hematopoietic stem cells and the expression dropped when the cells were induced to proliferate in culture [30,31,32]. Further characterization is required to delineate if these cells in the inner TZ are related to the hematopoietic or keratocyte lineage or represent a unique progenitor population.

Human CECs lack regenerative capacity in vivo, due to contact inhibition, expression of negative cell growth regulators, and presence of mitogenic inhibitors in aqueous humor [11]. This microenvironment could also cause the dormancy of TZ progenitors. Further study of the interplay between these progenitors and their niche, as well as their interaction with early differentiated cells in PE will be important to understand the regulatory identity in activating these endogenous cells in vivo and in vitro. Similar to other stem cell niches, the alteration or disruption of TZ niches might trigger the progenitors to respond and change [33]. An early study of selective argon laser trabeculoplasty, close to Schwalbe’s line, described uncontrolled endothelial cell growth and development of a cellular sheet extending from the region of Schwalbe’s line over the anterior TM surface [34]. This possibly disrupted the progenitor niche and caused the repair-type outgrowth to occlude the trabecular space, leading to a failure of trabeculoplasty.

The inner TZ and PE border as the progenitor niche was further explored under TEM and 3D-EM analyses. Inside the superficial stromal regions of inner TZ, cells were located in a loosely arranged matrix, fenestrated with numerous interstitial spaces, which might facilitate nutrient flow in maintaining the resident cells [35]. In contrast, the outer TZ stroma had more closely packed ECM interspersed with a mixture of pigmented and non-pigmented cells. This structure did not resemble the typical TM stroma, of which the fenestrated ECM beams and sheets are serially aligned to facilitate the aqueous humor flow [36]. Since there was no data of intraocular pressure (IOP) for the donor corneas provided by eye banks, we could not determine if the outer TZ stromal structure was affected by IOP changes or other ocular factors. Using 3D-EM (a combination of SBF-SEM and 3D reconstruction) to examine TZ/PE junction, we detected that the DM terminus was inserted beneath the TZ surface as far as one-third to one-half of the entire TZ width, and this resembled the breadth of inner TZ. Immunostaining of the human posterior limbal sections has revealed that this region beneath the TZ surface harbored cells expressing stem/progenitor and POM markers. Hence, it is probable that the DM terminus is part of the progenitor niche and might act as a physio-chemotactic guide to direct cells to move from the inner TZ to PE. Further cell tracing studies will help in delineating how these TZ progenitors link to PE cell clusters and will provide further information on niche and CE progenitors.

Ex vivo isolation and cultivation of human TZ progenitors, and whether the cells could differentiate to generate CE monolayers are under exploration. As donor eyes are cryo-preserved in Optisol for transportation, which usually takes a few days to over a week (corneas in this study had a mean Optisol preservation time of 11.5 days), and with unknown effects of Optisol reagent and hypothermic storage on progenitor viability, the harvest of viable TZ cells has been difficult to be standardized. Hence, we utilized fresh porcine eyes to establish primary TZ cell culture and differentiation, under a dual media approach [22]. Intriguingly, the TZ cells were able to generate confluent monolayer of endothelial-like cells with characteristic expression of pump-associated Na^+^K^+^ATPase and junctional complex ZO1 on the cell surface. Our study was limited by a lack of functional assays to validate if the differentiated cells behave the same as mature CECs. Further in vitro study of pump function using Ussing chamber system and in vivo implantation via cell injection or tissue-engineered endothelial keratoplasty to animal models of endothelial keratopathies will be important to elucidate the cell functions.

In conclusion, human TZ contained two phenotypically distinct regions: The inner and outer TZ. The inner TZ was enriched with progenitors expressing stem/progenitor markers. These progenitors projected as multicellular clusters into PE. The terminus of DM inserted beneath the inner TZ surface and, together with inner TZ, could represent the CE progenitor niche. The differentiation of porcine TZ cell into a CE-like monolayer has also verified the regenerative capacity of TZ cells. Further study of differentiating human TZ cells into a functional CE will demonstrate the translational potential for corneal endothelial regenerative therapy.

## Figures and Tables

**Figure 1 cells-08-01244-f001:**
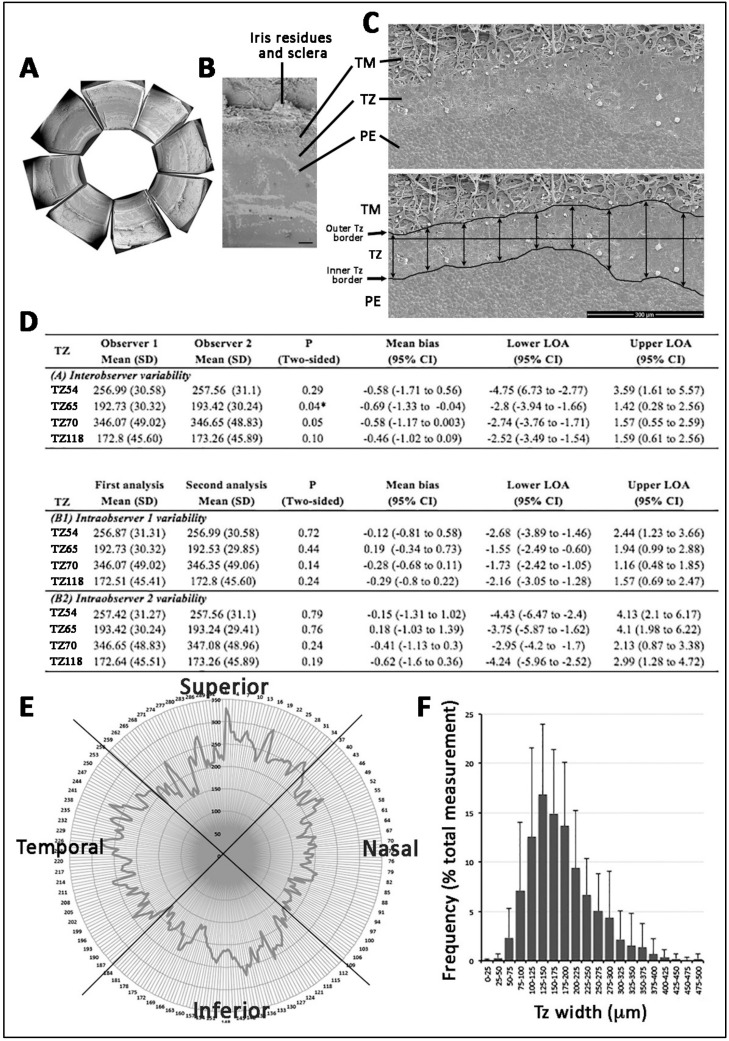
Human transition zone (TZ) morphology under scanning electron microscopy and TZ width profile. (**A**) Corneal periphery cut into equal pieces. (**B**) A magnified view showing regions of trabecular meshwork (TM), TZ and peripheral endothelium (PE). (**C**) Flat-mount view at 300× magnification and TZ outline with posterior border at insertions of uveal bands into TM and anterior border of polygonal-shaped endothelial cells. At every 100 μm interval, a vertical line was made between the inner and outer borders for TZ width measurement. (**D**) Inter- and intra-observer agreement of TZ width measurement. LOA: level of agreement. (**E**) 360° TZ width profile with respect to eye orientation. Nasal quadrant had the narrowest TZ. (**F**) A frequency plot of TZ width distribution. A dominant TZ width at 125 to 150 μm (15% coverage of overall TZ widths). Scale bar: 300 μm (B and C).

**Figure 2 cells-08-01244-f002:**
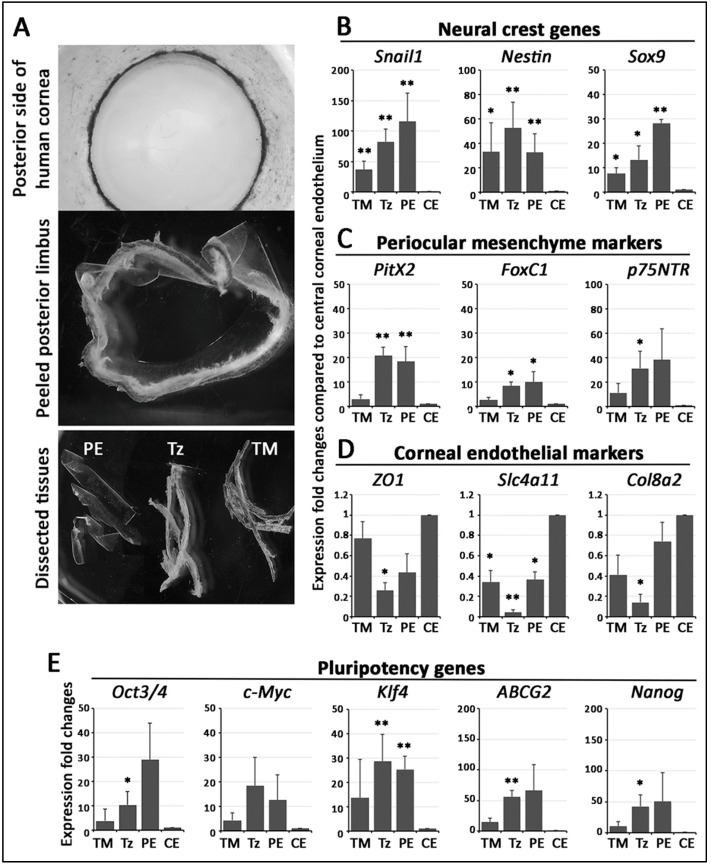
RNA expression in isolated human posterior limbus tissue. (**A**) Microdissection of posterior limbus into trabecular meshwork (TM), transition zone (TZ) and peripheral endothelium (PE) tissues for total RNA extraction. Target gene expression in various tissues compared to central corneal endothelium (CE). Data are presented as mean expression fold changes and SD. (**B**) Neural crest markers (*Snail1*, *Nestin*, *Sox9*). (**C**) POM markers (*Pitx2*, *FoxC1*, *p75^NTR^*). (**D**) Corneal endothelial differentiation marker (*ZO1*, *Slc4a11*, *Col8a2*). (**E**) Pluripotency markers (*Oct3/4*, *c-Myc*, *Klf4*, *ABCG2*, *Nanog*). * *p* < 0.05, ** *p* < 0.01, when compared to CE (Mann–Whitney U test).

**Figure 3 cells-08-01244-f003:**
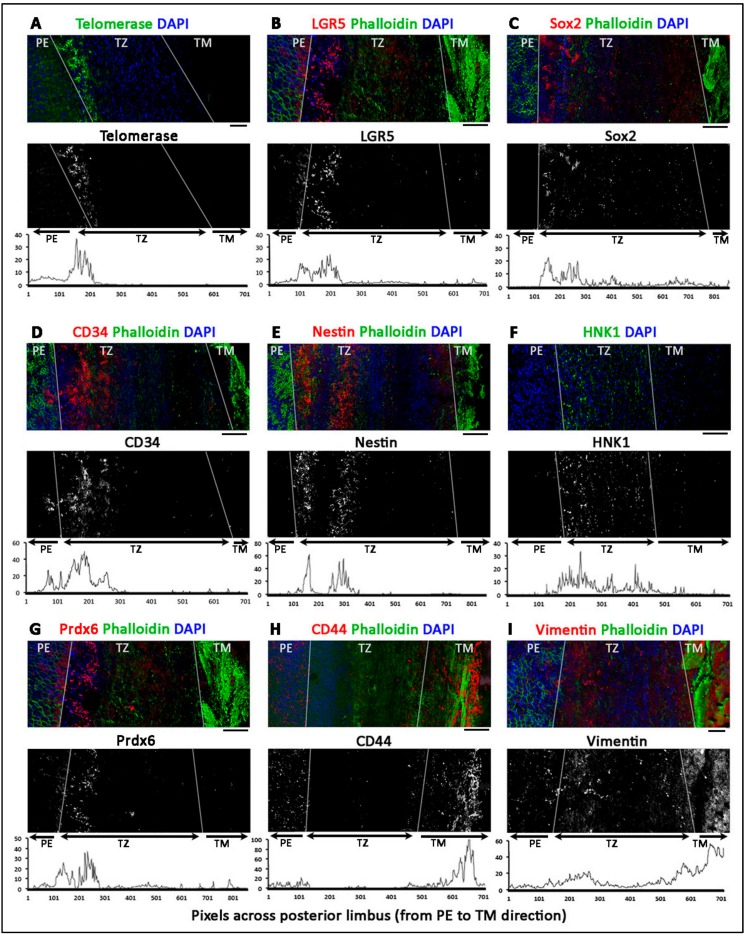
Z-series reconstructed confocal images showing the expression of stem cell markers in human transition zone (TZ), peripheral endothelium (PE) and trabecular meshwork (TM). The expression of (**A**) telomerase (TERT), (**B**) Lgr5, (**C**) Sox2, (**D**) CD34, (**E**) nestin and (**G**) Prdx6 were more expressed in the inner TZ. (**F**) HNK1 and (**I**) vimentin were detected in the entire TZ region. In contrast, outer TZ and TM had (**H**) CD44 and vimentin signal. White lines delineate borders between TM, TZ and PE. The top images represent the original colored confocal pictures, the middle grey-scaled images are single antibody-specific channel after threshold adjustment and the bottom graphs are the signal intensity profiles from PE across TZ to TM (*x*-axis represents distance in pixels and *y*-axis is the relative intensity levels). Scale bars: 50 μm.

**Figure 4 cells-08-01244-f004:**
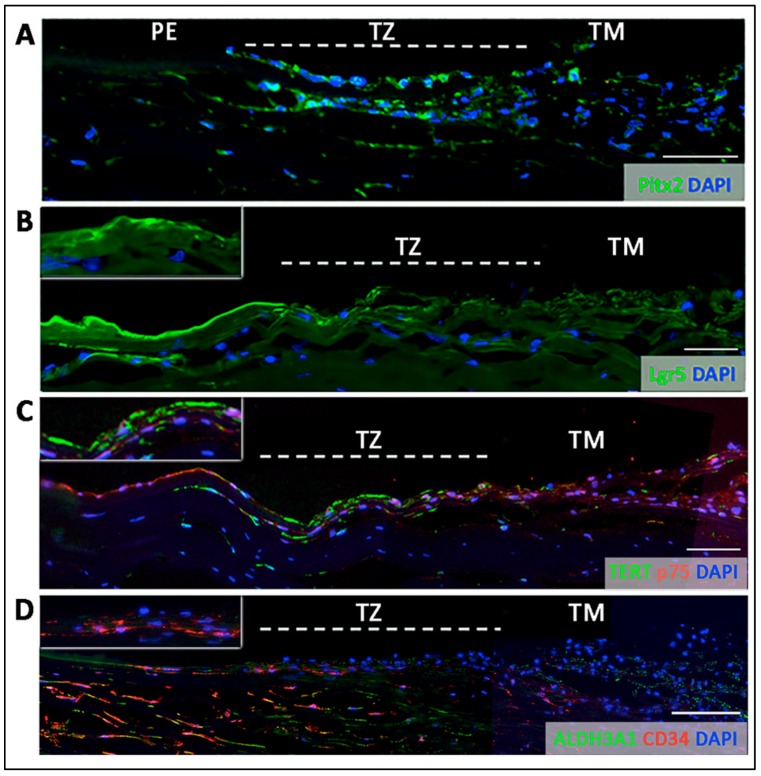
Immunostaining on transverse sections showing the expression of (**A**) nuclear Pitx2 across the entire transition zone (TZ). The stem cell markers, (**B**) Lgr5 on cell surface, (**C**) telomerase (TERT) and (**D**) CD34 were more expressed in the inner TZ. The positive cells were also located in the stroma immediately beneath the TZ surface (insets). (**D**) Quiescent stromal keratocytes co-expressed CD34 and aldehyde dehydrogenase 3A1 (ALDH3A1, a keratocyte marker). White dotted lines delineate TZ location between peripheral endothelium (PE) and trabecular meshwork (TM). Scale bars: 50 μm.

**Figure 5 cells-08-01244-f005:**
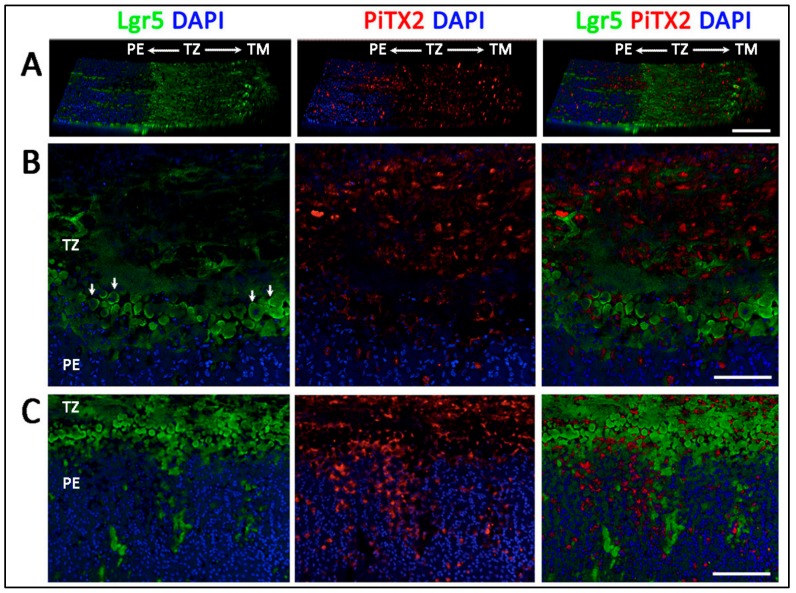
Immunolocalization of Lgr5 and Pitx2 in inner transition zone (TZ) and peripheral endothelium (PE) regions. (**A**) A horizontal view tilted at 30° showed Lgr5 predominantly localized in inner TZ while Pitx2 staining had a pan-TZ pattern. (**B**) Lgr5 and Pitx2 expression in inner TZ region, next to PE. (**C**) Lgr5 positive cell clusters extended from inner TZ into PE. Scale bars: 100 μm.

**Figure 6 cells-08-01244-f006:**
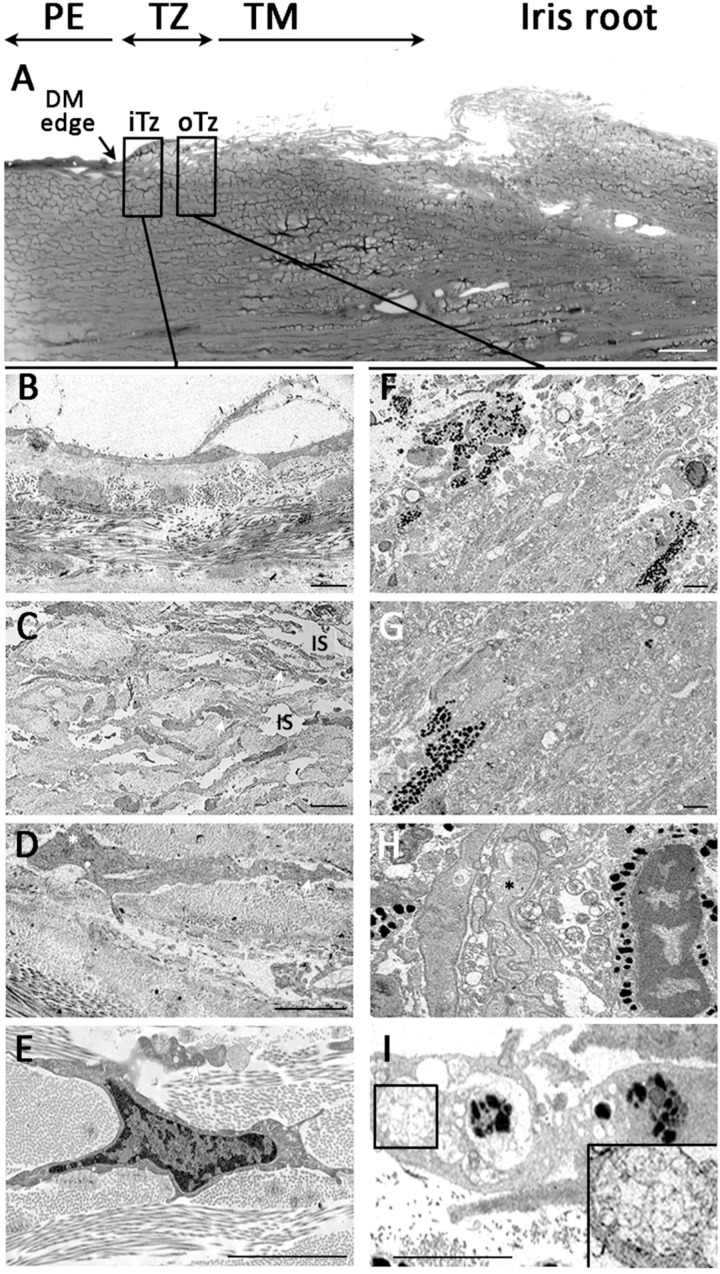
Ultrastructural morphology under transmission electron microscopy. (**A**) Overview picture illustrates posterior limbus with peripheral endothelium (PE) with DM edge, transition zone with inner and outer TZ, followed by trabecular meshwork (TM) and iris root. **Inner TZ.** (**B,C**) Cells in the loosely arranged stromal matrix with numerous interstitial spaces (IS). (**D,E**) The cells had high nuclear/cytoplasmic ratio and with loose chromatin and pronounced nucleoli (white asterisk in C). **Outer TZ.** (**F,G**) A mix of non-pigmented and pigmented cells in a closely packed matrix. (**H**) Cells containing cytoplasmic granules with pigmentation adjacent to highly convoluted non-pigmented cells. (**I**) Cell with phagosomes containing irregular deposits (magnified image in inset). Abbreviations: iTZ: inner TZ; oTZ: outer TZ; DM: Descemet’s membrane. Scale bars: 50 μm (A); 10 μm (B–F, I); 5 μm (G).

**Figure 7 cells-08-01244-f007:**
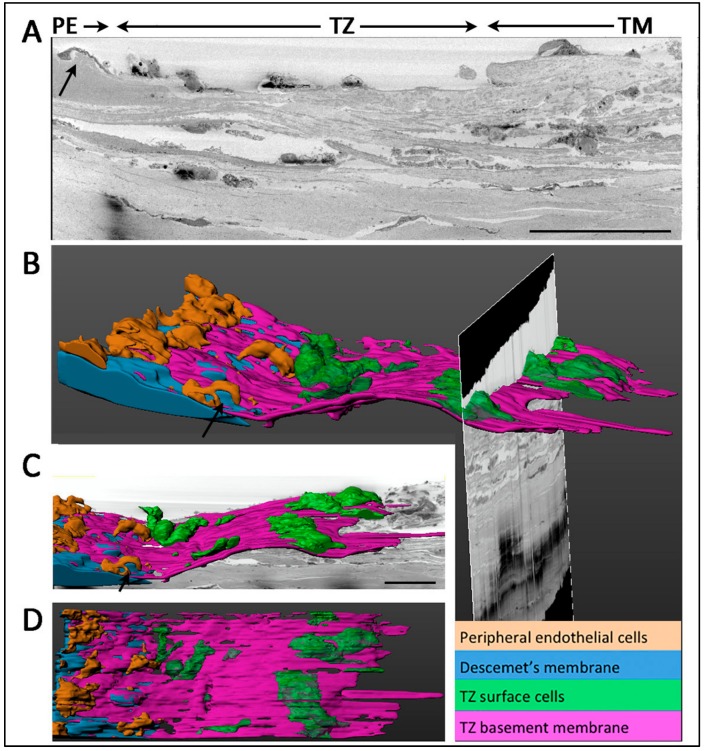
Serial block face-scanning electron microscopy of junction between transition zone (TZ) and peripheral endothelium (PE) and 3D reconstruction. (**A**) Representative TEM slice showing an overview of PE/TZ junction. Arrows show the presence of Hassall-Henle structure at extreme peripheral Descemet’s membrane (DM). (**B**) 3D reconstructed image of TZ/PE junction showing DM insertion below TZ surface. (**C**) 3D reconstructed image showing TZ surface cells. (**D**) En face view showing the distribution of PE and TZ surface cells. Scale bars: 50 μm.

**Figure 8 cells-08-01244-f008:**
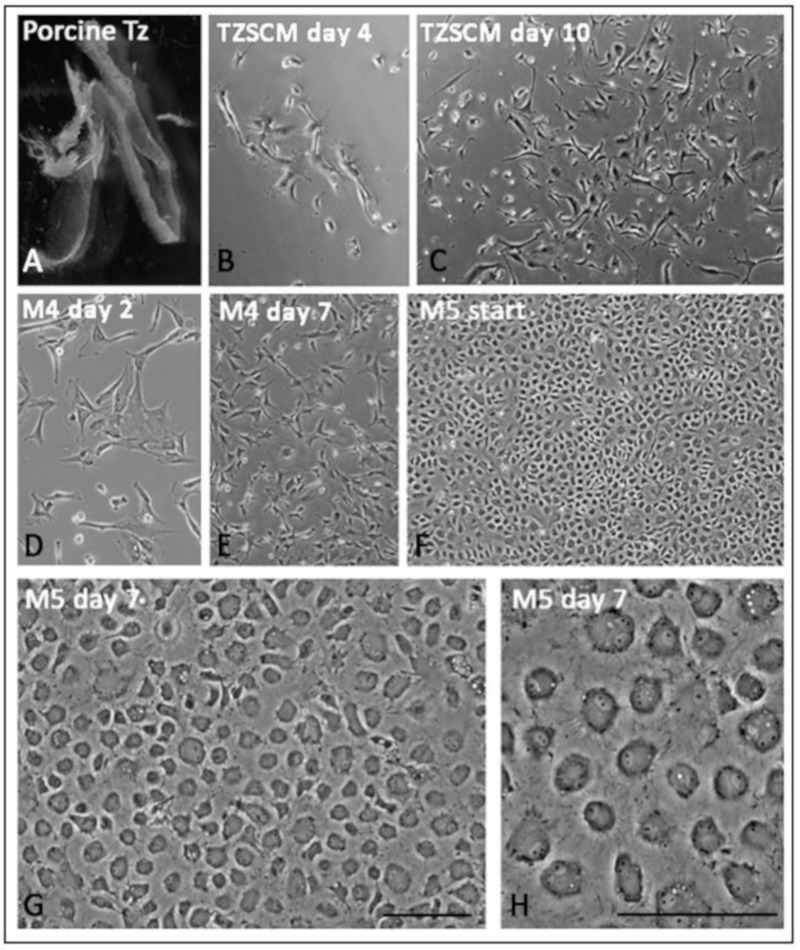
Primary culture of porcine transition zone (TZ) cells. (**A**) Dissected TZ tissue with minimal TM and PE. (**B**,**C**) Primary TZ cells in culture with TZ stem cell medium (TZSCM) for 4 and 10 days, respectively. (**D–F**) Primary TZ cells in proliferative M4 medium at day 2, 7 and 14 for monolayer formation. (**G,H**) Cells in endothelial stabilizing M5 medium for 7 days. A homogenous monolayer of tightly-packed polygonal to hexagonal-like cells was generated. Scale bars: 50 μm.

**Figure 9 cells-08-01244-f009:**
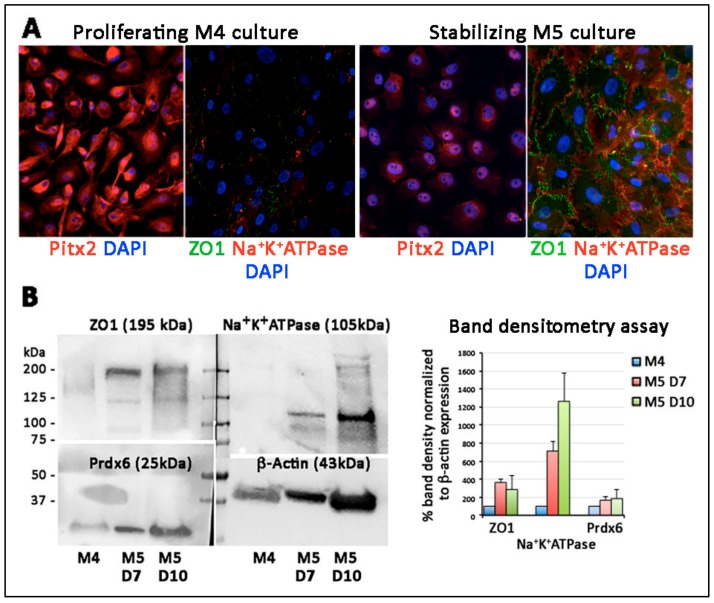
Characterization of porcine transition zone (TZ)-generated endothelial-like cells. (**A**) Immunostaining of POM marker (Pitx2) and corneal endothelial differentiation markers (ZO1, Na^+^K^+^ATPase) in M4 proliferative and M5 stabilization cultures. The expression of ZO1 and Na^+^K^+^ATPase was detected in M5 cultured cells, which had reduced Pitx2 expression. (**B**) Western blotting of ZO1, Na^+^K^+^ATPase and Prdx6. The nitrocellulose blot was cut with reference to the molecular weight range for primary antibody incubation and direct comparison of marker expression. Band densitometry analysis after normalization of β-actin expression showed elevated ZO1 and Na^+^K^+^ATPase expression under M5 stabilization culture for 7 and 10 days, respectively.

**Table 1 cells-08-01244-t001:** Site-specific expression of stem/progenitor and corneal endothelial differentiation markers.

Posterior Limbus Regions	Marker Expression
Peripheral endothelium (PE)	Lgr5, Prdx6, nestin (sporadic), CD34 (scarce), TERT (sporadic), ZO1, Na^+^K^+^ATPase
Transition zone (TZ) Entire TZ	Vimentin, HNK1, Pitx2, p75^NTR^
Inner TZ	Lgr5, Prdx6, CD34, TERT, nestin
Outer TZ	CD44
Trabecular meshwork (TM)	Vimentin, CD44

Note: cells immunopositive to Lgr5, TERT, CD34, Pitx2 were detected in posterior stromal region close to the inner TZ surface.

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
