# Peer review of "Characterization of Human Transition Zone Reveals a Putative Progenitor-Enriched Niche of Corneal Endothelium"

_cells, 2019, doi:10.3390/cells8101244_

Round 1

Reviewer 1 Report

This paper presented some findings about the characterization of human transition zonein corneal endotelium. However, this paper is not acceptable at present, unless some fundamental problems are settled.

Major concern

The patholphysiological role of TZ cellsprovided by the study are limited. The authors need to address this issue in more detail.

2. Because the resolution of images in Fig 3 and 5 is low. I could not conclude your results. The sacle bars were missing in these images.

Posterior limval tissues were seperated into TM, TZ, PE and central CE. How did author identify these tissue samples exactly? Are There specific tissue markers for TM, TZ, PE? How could you determine the contamination of other tissue parts?

4. Author used human corneal tissues. Author have to provide ethecal statement and institutional approval number.

Author Response

This paper presented some findings about the characterization of human transition zonein corneal endotelium. However, this paper is not acceptable at present, unless some fundamental problems are settled.

Reply – Thank you for reviewing our work. We have substantially revised the manuscript according to comments and hope it can be accepted for publication.

Major concern

The patholphysiological role of TZ cellsprovided by the study are limited. The authors need to address this issue in more detail.

Reply – Although a few studies have reported TZ in the corneal periphery, it is still poorly characterized. Previous studies using stem cell marker expression and sphere culture have suggested the existence of undifferentiated cells with proliferation potential in the corneal periphery, however the exact location is unclear. Hence, we characterized the human TZ using various methods from gene expression to anatomical studies with different microscopy methods to identify the location of progenitors and their niche. This point has been stated in Introduction (line 69-82). Also, due to presence of various factors in the anterior chamber and aqueous humor to inhibit cell proliferation, TZ progenitors could be kept in dormancy under normal or even in disease situations. Hence, the pathophysiological role of these cells is currently unknown. Our group is working on the isolation and culture of human TZ cells, and will study the differentiation potential to mature CEC. We will examine the growth efficiency and CEC differentiation capacity of TZ cells isolated from normal corneas as well as corneas with known endothelial diseases to elucidate any physiological impacts due to pathological conditions.

Because the resolution of images in Fig 3 and 5 is low. I could not conclude your results. The sacle bars were missing in these images.

Reply – We have revised Figure 3 using high-resolution images with proper labeling and added the intensity profiles to illustrate the expression intensity changes of different markers across PE, TZ and TM regions. The immunostaining on transverse sections have been placed in a new Figure 4. The other figures are re-numbered accordingly. Fig. 6 (original Fig. 5) showing TEM results of inner and outer TZ was revised and the structures/regions of interest have been highlighted by arrows and asterisks. All figures have been added with appropriate scale bars.

Posterior limval tissues were seperated into TM, TZ, PE and central CE. How did author identify these tissue samples exactly? Are There specific tissue markers for TM, TZ, PE? How could you determine the contamination of other tissue parts?

Reply – We harvested various tissue parts sequentially from human corneas. The central CE was firstly collected by central DM peeling (8 mm diameter). The posterior limbus was then isolated by an inward peeling method (a video clip from youtube was referenced in the paper). Under stereomicroscopy, the PE was separated as a transparent tissue of <0.5 mm width and TM was the outermost pigmented tissue. The remaining non-pigmented part was the smooth TZ. The isolated tissues were washed thrice in ice-cold PBS and carefully checked to ensure an optimal clearance of other tissue parts, before RNA extraction. This point has been added in Materials and Methods (line 133-138).

Author used human corneal tissues. Author have to provide ethecal statement and institutional approval number. 

Reply – The information has been added in Materials and Methods (line 88-91).  

Reviewer 2 Report

In ‘Characterization of Human Transition Zone Reveals a Putative Progenitor-Enriched Niche of Corneal Endothelium’, Yam et al. report on a population of cells expressing stem/progenitor markers such as ABCG2, Lgr5, Pitx2, and telomerase found in the human inner transition zone of the posterior limbus at the corneal periphery. They further isolated similar porcine cells, which differentiated and expressed corneal endothelial markers. This introduction situates well the research question and highlights the current gap in the literature (insufficient characterization of the transition zone), and the materials and methods are clearly written and provided in detail. Overall, the study seems well designed and the results are clearly presented. I do have a few comments/questions:

The corneal samples were prepared for the SEM imaging and transition zone measurements by dehydration, among other processing steps. Could this lead to artifacts (shrinkage)? The authors criticized other studies regarding fixation techniques (lines 423-425) but don’t consider this as a limitation of their own work. In the sections ‘Phenotypically distinct regions inside TZ’ and ‘Projection of progenitors from transition zone to peripheral endothelium’ as well as in Figures 3 and 4, it is unclear how the trabecular meshwork, transition zone, and peripheral endothelium areas are identified. This is clearly described and presented for the SEM images and measurements of transition zone width but not for the immunohistochemistry images. Figure 5 is unclearly presented. First, in the top panel, the orientation of the tissue is unclear. How are the inner and outer transition zones oriented relative to the trabecular meshwork and the peripheral endothelium. Next, the labeling of the columns with A and B seems redundant since the individual images are labeled. Additionally, scale bars are missing. Finally, it would be helpful to the reader to indicate key features with arrows (e.g., the lamellated deposits). The porcine cells were cultured in an endothelial stabilizing M5 medium (Human Endothelial-SFM with 5% serum)? What is the composition of this medium and who is the supplier? In the discussion, the authors claim that this was a natural differentiation without any special medium (lines 511-513); however, I would assume that this Endothelial SFM might contain some pro-angiogenic factors.

Author Response

In ‘Characterization of Human Transition Zone Reveals a Putative Progenitor-Enriched Niche of Corneal Endothelium’, Yam et al. report on a population of cells expressing stem/progenitor markers such as ABCG2, Lgr5, Pitx2, and telomerase found in the human inner transition zone of the posterior limbus at the corneal periphery. They further isolated similar porcine cells, which differentiated and expressed corneal endothelial markers. This introduction situates well the research question and highlights the current gap in the literature (insufficient characterization of the transition zone), and the materials and methods are clearly written and provided in detail. Overall, the study seems well designed and the results are clearly presented. I do have a few comments/questions:

The corneal samples were prepared for the SEM imaging and transition zone measurements by dehydration, among other processing steps. Could this lead to artifacts (shrinkage)? The authors criticized other studies regarding fixation techniques (lines 423-425) but don’t consider this as a limitation of their own work.

Reply Thank you for the comment. We performed standard SEM protocol of tissue fixation by glutaldehyde and OsO4, followed by dehydration with ascending alcohol series. The samples then underwent critical point drying with CO2 as transitional fluid, to avoid tissue shrinkage and artifacts. This point has been stated in Materials and Methods (line 115-120) and in Discussion (line 448-450).

In the sections ‘Phenotypically distinct regions inside TZ’ and ‘Projection of progenitors from transition zone to peripheral endothelium’ as well as in Figures 3 and 4, it is unclear how the trabecular meshwork, transition zone, and peripheral endothelium areas are identified. This is clearly described and presented for the SEM images and measurements of transition zone width but not for the immunohistochemistry images.

Reply – For the expression of various markers, we performed co-staining with phalloidin and DAPI to facilitate the identification of PE, TZ and TM. PE was identified with tightly packed DAPI-stained nuclei whereas TM had strong phalloidin staining due to the extensive fibrillar insertions and ridges. In contrast, TZ was distinguished by the relatively dim signals of DAPI and phalloidin. This point has been added in Results (line 277-280).

Figure 5 is unclearly presented. First, in the top panel, the orientation of the tissue is unclear. How are the inner and outer transition zones oriented relative to the trabecular meshwork and the peripheral endothelium. Next, the labeling of the columns with A and B seems redundant since the individual images are labeled. Additionally, scale bars are missing. Finally, it would be helpful to the reader to indicate key features with arrows (e.g., the lamellated deposits).

Reply – Thank you for the comment and suggestion. The TEM results in revised Fig. 6 contain clear labeling of orientation from PE to TM and iris root. We have highlighted the structures/regions of interest using arrows and asterisks and added scale bars.

The porcine cells were cultured in an endothelial stabilizing M5 medium (Human Endothelial-SFM with 5% serum)? What is the composition of this medium and who is the supplier? In the discussion, the authors claim that this was a natural differentiation without any special medium (lines 511-513); however, I would assume that this Endothelial SFM might contain some pro-angiogenic factors.

Reply – Thank you for the comment. We used Human Endothelial Serum-free medium supplemented with b-FGF, EGF, human plasma fibronectin and Equafetal bovine serum for the porcine TZ cell culture. The detailed formulation has been added in Materials and Methods (line 179-182). Also the statement of natural differentiation has been removed.    

Reviewer 3 Report

The authors present a thorough morphological characterisation of the marginal posterior corneal zone, including peripheral endothelium, transition zone and beginning trabecular meshwork, with a focus on putative stem cell identification and niche localisation. They use a broad panel of methods such as (immuno)histochemistry, various electron microscopic techniques, PCR, western blotting, and cell culture. The study is sound and the methods are adequate. The authors present novel and important findings regarding the putative posterior corneal stem cell niche and stem cells residing therein. The manuscript is well written and comprehensible, however, some revision would improve the manuscript, see following detailed comments:

Methods page 3 line 105 and following, length specifications: What character is this curl appearing in most of the dimensions, is it due to a false key or font style, or maybe file conversion? Methods page 3 line 111: There can only be 4 quadrants and not 8, but there can be 8 pieces. Please correct. Results supplemental figure 4 legend: Please assign respective EM mode to the insert magnifications of each region to help the interested reader understand the figure. Results immunostainings: Please provide negative and if available also positive control stainings for each antibody. Results page 8 line 270, figure 3, table 1: How do the authors explain or interpret the finding of CD34 positivity and gene expression in the inner TZ/PE? CD34 is a marker for hematopoietic stem cells and vascular endothelial progenitors. One would expect to find CD34 positivity at the TM and Schlemm´s canal but not at or adjacent to the corneal endothelium. Results figure 4: Please indicate directions allowing for 3D recognition. So far it appears as 2D only and directions are unclear, for example, what direction is indicated with “horizontal” and which direction is indicated with “en face”? Maybe also insert an explanatory scheme? This would be of great help for the interested reader, especially for those who have limited EM experience. Results figure 5, page 12 line 327: It would be helpful to indicate cellular structures on the EM pictures. A “loss of chromatin” would indicate apoptotic DNA degradation. Did the authors perhaps mean “loose chromatin” (euchromatin)? Results, general, terminology: I know that it has become common practice to speak of expression and up-/down-regulation irrespective of whether genes or proteins are the subject. Nonetheless I would like to suggest that authors consider proper terminology (genes are expressed and can be up- or down-regulated, while proteins are produced and their levels increase or decrease). Discussion, general: The discussion section contains substantial results repetitions and should be shortened to restrict it to data interpretation and discussion. For example, the first four paragraphs of the discussion section (page 15 line 393 to page 16 line 439) contain a lot of descriptive and repetitive elements of the results section, but interpret and discuss results only marginally. Furthermore, it is mentioned several times that these observations are published “for the first time”, which is not necessary because peers in the research community are able to recognize novel findings without being told. Discussion page 16 lines 437-439: After describing that the TZ is narrower at the nasal quadrant the authors mention that certain disease conditions like pinguecula or pterygium occur mainly nasally. This provokes speculations about a link between these diseases and the posterior nasal limbus, but the conditions mentioned are diseases of the conjunctiva and a relation between the TZ and the conjunctival function at the nasal side is not supported by data so far. I would suggest to omit this sentence. Discussion page 17 lines 484-487: The authors did not prove by staining or any other method that the cells they observed are dendritic cells, i.e. part of the cellular immune system. It appears also highly speculative to conclude that these cells and the architecture of this region may be responsible for coordination of cellular activities as response to metabolic changes. Further data is needed to address this aspect with regard to the posterior corneal limbus.

Author Response

The authors present a thorough morphological characterisation of the marginal posterior corneal zone, including peripheral endothelium, transition zone and beginning trabecular meshwork, with a focus on putative stem cell identification and niche localisation. They use a broad panel of methods such as (immuno)histochemistry, various electron microscopic techniques, PCR, western blotting, and cell culture. The study is sound and the methods are adequate. The authors present novel and important findings regarding the putative posterior corneal stem cell niche and stem cells residing therein. The manuscript is well written and comprehensible, however, some revision would improve the manuscript, see following detailed comments:

Methods page 3 line 105 and following, length specifications: What character is this curl appearing in most of the dimensions, is it due to a false key or font style, or maybe file conversion?

Reply – We have corrected these typos, which were generated when the word document file was converted to the journal’s peer-review format. All corrections have been highlighted.

Methods page 3 line 111: There can only be 4 quadrants and not 8, but there can be 8 pieces. Please correct.

Reply – Thank you for the correction. We have revised accordingly.

Results supplemental figure 4 legend: Please assign respective EM mode to the insert magnifications of each region to help the interested reader understand the figure.

Reply – Thank you. We have highlighted the pictures taken under the back-scattered electron mode with bold white frame. This should help the readers visualize the electron dense gold particles. Scale bars have also been added.

Results immunostainings: Please provide negative and if available also positive control stainings for each antibody.

Reply - Results of immunostaining using IgG isotype antibody as negative control has been added in Supplementary Fig. S4. This has been mentioned in Results (line 283-284).

Results page 8 line 270, figure 3, table 1: How do the authors explain or interpret the finding of CD34 positivity and gene expression in the inner TZ/PE? CD34 is a marker for hematopoietic stem cells and vascular endothelial progenitors. One would expect to find CD34 positivity at the TM and Schlemm´s canal but not at or adjacent to the corneal endothelium.

Reply – A review from Sidney et al. (Stem Cells 2014) has illustrated that CD34 is a marker of diverse progenitors, including hematopoietic and endothelial progenitors as well as non-hematopoietic stem cells (such as epithelial progenitors, corneal stromal stem cells) and quiescent corneal stromal keratocytes (CSK). This supports our immunostaining results that some inner TZ surface cells and cells immediately beneath the TZ surface were CD34 positive. These TZ cells were different from CSK, which co-expressed both CD34 and ALDH3A1 (a keratocyte marker) (Fig 4D). This point has been added in Results (line 303-306).

Results figure 4: Please indicate directions allowing for 3D recognition. So far it appears as 2D only and directions are unclear, for example, what direction is indicated with “horizontal” and which direction is indicated with “en face”? Maybe also insert an explanatory scheme? This would be of great help for the interested reader, especially for those who have limited EM experience.

Reply – Thank you for the comments. We have removed “3D” and “en face” to avoid confusion. The horizontal view was obtained with a tilted 30° angle, for visualizing the staining signal across the landscape of PE, TZ and TM. Revision has been made in Results (line 334, 345).

Results figure 5, page 12 line 327: It would be helpful to indicate cellular structures on the EM pictures. A “loss of chromatin” would indicate apoptotic DNA degradation. Did the authors perhaps mean “loose chromatin” (euchromatin)?

Reply – Thank you for the suggestion. We have revised the TEM pictures in new Fig. 6 with labelings of structures of interest. We also corrected to “loose chromatin” for inner TZ cells (line 355).

Results, general, terminology: I know that it has become common practice to speak of expression and up-/down-regulation irrespective of whether genes or proteins are the subject. Nonetheless I would like to suggest that authors consider proper terminology (genes are expressed and can be up- or down-regulated, while proteins are produced and their levels increase or decrease).

Reply – Thank you for the comments. We have checked through the manuscript for correct terminology.

Discussion, general: The discussion section contains substantial results repetitions and should be shortened to restrict it to data interpretation and discussion. For example, the first four paragraphs of the discussion section (page 15 line 393 to page 16 line 439) contain a lot of descriptive and repetitive elements of the results section, but interpret and discuss results only marginally.

Reply - The Discussion has been revised and repetitions were removed. All changes are highlighted in red.

Furthermore, it is mentioned several times that these observations are published “for the first time”, which is not necessary because peers in the research community are able to recognize novel findings without being told.

Reply – This phase has been removed.

Discussion page 16 lines 437-439: After describing that the TZ is narrower at the nasal quadrant the authors mention that certain disease conditions like pinguecula or pterygium occur mainly nasally. This provokes speculations about a link between these diseases and the posterior nasal limbus, but the conditions mentioned are diseases of the conjunctiva and a relation between the TZ and the conjunctival function at the nasal side is not supported by data so far. I would suggest to omit this sentence.

Reply – Thank you for the comments. We agree that it is inappropriate to explain phenomenon found in normal tissue by quoting disease conditions. We have removed this point from the Discussion.

Discussion page 17 lines 484-487: The authors did not prove by staining or any other method that the cells they observed are dendritic cells, i.e. part of the cellular immune system. It appears also highly speculative to conclude that these cells and the architecture of this region may be responsible for coordination of cellular activities as response to metabolic changes. Further data is needed to address this aspect with regard to the posterior corneal limbus.

Reply – We have removed “dendritic” to avoid confusion of describing cells with extensive cell processes (line 356, 366, 493). In addition, the sentence related to interstitial spaces in inner TZ has been revised to possibly facilitating nutrient flow in maintaining the resident cells (line 494-495). This avoids the speculation on tissue functions without supporting data.

Reviewer 4 Report

Corneal endothelium is a vital part for functional activity of the cornea and in human it is believe that endothelium do not proliferate which is unlike many other animal species. As human endothelium do not undergo proliferation or as per the author of this journal, lack the regenerative capacity, to find out stem cell source of the endothelium is a prime research interest. Despite tremendous effort, few report claim the presence of endothelial progenitor cells in the eye, none of them have any real implementation for the treatment of the diseased patients. The current study reports the identification of Transition Zone containing progenitor-like cells, which can serve for the regeneration of the corneal endothelium. Overall, the manuscript is well written. However, I am concerned about the novelty of this work. There are published reports saying that progenitor cells in the Transition Zone can serve to replace the endothelium even in the disease condition, which is same as the claim of this current paper. Marker expression which is presented in the paper already shown and overall only marker expression does not prove the functional acuity of the cells. Electron microscopy was also studied. Studies with porcine eyes could have been interesting with human eyes. Functionality test is very important. Lack of in vivo validation is also a limitation of this study.

Author Response

Corneal endothelium is a vital part for functional activity of the cornea and in human it is believe that endothelium do not proliferate which is unlike many other animal species. As human endothelium do not undergo proliferation or as per the author of this journal, lack the regenerative capacity, to find out stem cell source of the endothelium is a prime research interest. Despite tremendous effort, few report claim the presence of endothelial progenitor cells in the eye, none of them have any real implementation for the treatment of the diseased patients. The current study reports the identification of Transition Zone containing progenitor-like cells, which can serve for the regeneration of the corneal endothelium. Overall, the manuscript is well written. However, I am concerned about the novelty of this work. There are published reports saying that progenitor cells in the Transition Zone can serve to replace the endothelium even in the disease condition, which is same as the claim of this current paper. Marker expression which is presented in the paper already shown and overall only marker expression does not prove the functional acuity of the cells. Electron microscopy was also studied. Studies with porcine eyes could have been interesting with human eyes. Functionality test is very important. Lack of in vivo validation is also a limitation of this study.

Reply – Thank you for reviewing our paper. Although a few studies have reported on the TZ in the corneal periphery, it has been poorly characterized, hence the purpose of this paper. Previous studies using stem cell marker expression and sphere culture have suggested the existence of undifferentiated cells with proliferation potential in the corneal periphery, however the exact location is unclear. Therefore, we characterized the human TZ using various methods from gene expression to anatomical studies with different microscopy methods to identify the location of progenitors and their niche. This point has been stated in Introduction (line 69-82). We also, for the first time, identified two distinct zones in the TZ region. Using porcine TZ cell culture and differentiation to endothelial-like cells, this is strong evidence of TZ potency in corneal endothelial regeneration. Our current work is focused on the isolation and culture of human TZ cells, and studying the differentiation potential to mature CEC. We will examine the growth efficiency and CEC differentiation capacity of TZ cells isolated from normal corneas to elucidate the translational potential for corneal endothelial regeneration. In the revised Discussion, we have added the limitation of this work with a lack of functional assays, such as in vitro pump function by Ussing chamber system and in vivo animal study (line 517-521).

Round 2

Reviewer 1 Report

The manuscript was improved. However, some main details should be clarified. Authors separated mechanically CE, PE and TM from the cornea using an inward peeling method. Isolated tissue parts were washed to clear other tissue parts. Do this mechanical peeling and washing method really guarantee the no contamination of other tissue parts? A gene in a single cell from the contaminated tissue part can be amplified by PCR, which will distort the final result. 

Author Response

Reply – Thank you for the comment. We have provided more detailed procedure of tissue isolation in Materials and Methods (line 135-138). We also added a discussion of the possible contamination of other tissue parts, hence we studied a larger number of samples to ensure the expression result was a general phenomenon among corneas (Discussion line 471-476).

Reviewer 3 Report

The authors have responded reasonably to most of my comments. Still, some questions remain open and should be answered by the authors:

The authors missed to add the requested images from negative controls to immunostainings, although they replied that images of IHC stainings with IgG isotype control were included in the supplemental file as supplementary figure S4. But supplementary figure S4 is showing the EM images, and IHC images are missing in the file. I assume that the authors wanted to add a supplementary figure S5? The authors underpinned their findings and statement on CD34 positivity as a marker for CSK with a reference (Sidney et al., ref. no. 24) that describes CD34 as a marker for a variety of stem and progenitor cells, including keratocytes. However, the referenced review article by Sidney et al. cites primary literature sources that clearly state that a subset of cells in the corneal stroma and/or the stromal limbus is CD34 positive and that these cells resemble stem cells, but are not keratocytes. In these primary references, the ones from own group, namely the group of Dua (Joseph et al. Invest Ophthalmol Vis Sci. 2003 Nov;44(11):4689-92; Perella et al., Br J Ophthalmol. 2007 Jan;91(1):94-9; Branch et al. Invest Ophthalmol Vis Sci. 2012 Aug 3;53(9):5109-16; Hashmani et al. Stem Cell Res Ther. 2013 Jun 24;4(3):75), report that CD34 is a keratocyte marker based on analyses of cell populations isolated from whole corneal stroma. However, the Perella, Branch and Hashmani papers contain more detailed analyses showing that only a subset of cells in the corneal stroma is CD34 positive and this subset consists of stem cells rather than  keratocytes. Hence the frequently used descriptive term of CD34 as a marker for keratocytes is confusing. Interestingly, the referenced primary literature by Sosnova et al. (Stem Cells. 2005 Apr;23(4):507-15) shows, albeit in rodent, that CD34 positive cells in the corneal stroma are derived from bone marrow and resemble hematopoietic stem cells and not keratocytes. Their findings are supported by the referenced primary studies of Polisetty et al. (Mol Vis. 2008; 14: 431–442) and Choong et al. (Cytotherapy. 2007;9(3):252-8), as examples. Having read all these papers I am not convinced that CD34 is a keratocyte marker and that CD34 positive cells in the TZ and beneath are CSK as the authors responded, and think this issue needs to be adressed in a more complex and detailed way. In the new figure 6h it is not possible to recognize the denoted structure as lamellated deposit due to the resolution of the image. If this is an important finding it should be displayed at higher magnification/resolution.

Author Response

1. We have re-uploaded the supplementary information with Fig S4 showing IHC controls. 

2. Thank you for the suggestion. We have added a discussion of CD34 expression in various progenitors and stromal keratocytes as well as the interpretation to our result of CD34 expression in inner TZ cells in Discussion (line 487-496). 

3. We have added an inset with magnified image to clearly illustrate the lamellated deposits (line 372).

Reviewer 4 Report

Thanks to the authors to acknowledge relevant published works in the introduction although its always good to see the original work rather than review. Thanks again to mentioned the limitation of the study in the discussion. 

Author Response

Thank you for the positive comments to our revised version. We have edited English and spelling check of the entire manuscript.

Round 3

Reviewer 1 Report

The manuscript was improved.

Author Response

Thank you for the comments. We have revised the Methods with more clear descriptions and included the proper control experiments for immunostaining. Also, we edited the English through the manuscript. All changes are tracked in red.  

Reviewer 3 Report

The authors have improved the manuscript, however, some minor issues remain:

The authors added the requested images from negative controls to immunostainings with an IgG control antibody, although they used mouse IgG, mouse IgM and rabbit polyclonal antibodies. What about controls for the IgM and the rabbit antibodies? The figure legend does not give any precise information on that. In the new figure 6h it is not possible to recognize the structure in the enlarged inset as "lamellated deposit", because the term "lamellated" implies a layered architecture and the deposit looks rather amorphous. Cells of the outer TZ and TM are frequently challenged with shed proteins that need to be cleared of, but these are usually not lamellated, so where could a lamellated structure be derived of? Since the authors do not interpret this finding I suggest to simply omit the somewhat misleading "lamellated".

Author Response

Thank you for the comments. We have revised Supplementary Fig. S4 with new pictures showing the immunostaining without primary antibody, with rabbit IgG, mouse IgG and IgM isotype antibodies. All showed negligible staining signals. Accordingly, we have revised the Materials and Methods (2nd paragraph, page 3), Results (page 8 to 9) and Supplementary information Fig. S4 and its legend, as well as in Table S2 (added a new column of antibody isotype).

We also replaced “lamellated deposits” with “irregular deposits” (on page 12 last line and Fig. 6 legend). In addition, Fig. 6 has been relabeled with upper case letters to make it consistent with other figures.